# Development of Optimized Vitrification Procedures Using Closed Carrier System to Improve the Survival and Developmental Competence of Vitrified Mouse Oocytes

**DOI:** 10.3390/cells10071670

**Published:** 2021-07-02

**Authors:** Jae Kyun Park, Ju Hee Lee, Eun A Park, Hyunjung J. Lim, Sang Woo Lyu, Woo Sik Lee, Jayeon Kim, Haengseok Song

**Affiliations:** 1Department of Biomedical Sciences, CHA University, Seongnam 13488, Korea; pjk987427@chamc.co.kr (J.K.P.); joohee8406@naver.com (J.H.L.); 2CHA Fertility Center Gangnam, CHA University, Seoul 06125, Korea; dung5038@cha.ac.kr (S.W.L.); wooslee@cha.ac.kr (W.S.L.); 3CHA Fertility Center Seoul Station, CHA University, Seoul 04637, Korea; eapark9327@chamc.co.kr; 4Department of Veterinary Medicine, School of Veterinary Medicine, Konkuk University, Seoul 05029, Korea; hlim@konkuk.ac.kr

**Keywords:** oocyte vitrification, closed carrier systems, cross-contamination, cooling rate, warming rate, exposure time to cryoprotectants, viability, developmental competence

## Abstract

The open carrier system (OC) is used for vitrification due to its high efficiency in preserving female fertility, but concerns remain that it bears possible risks of cross-contamination. Closed carrier systems (CC) could be an alternative to the OC to increase safety. However, the viability and developmental competence of vitrified/warmed (VW) oocytes using the CC were significantly lower than with OC. We aimed to improve the efficiency of the CC. Metaphase II oocytes were collected from mice after superovulation and subjected to in vitro fertilization after vitrification/warming. Increasing the cooling/warming rate and exposure time to cryoprotectants as key parameters for the CC effectively improved the survival rate and developmental competence of VW oocytes. When all the conditions that improved the outcomes were applied to the conventional CC, hereafter named the modified vitrification/warming procedure using CC (mVW-CC), the viability and developmental competence of VW oocytes were significantly improved as compared to those of VW oocytes in the CC. Furthermore, mVW-CC increased the spindle normality of VW oocytes, as well as the cell number of blastocysts developed from VW oocytes. Collectively, our mVW-CC optimized for mouse oocytes can be utilized for humans without concerns regarding possible cross-contamination during vitrification in the future.

## 1. Introduction

Considering an increase in cancer survival rates, especially in young patients, the preservation of fertility is critical [1]. Fertility preservation among these patients is preferred [2,3], resulting in successful childbirth [4,5]. Moreover, the benefits of oocyte cryopreservation are not only limited to cancer patients, but also extend to women who want to delay childbirth [6,7]. Thus, oocyte cryopreservation for medical or social reasons is considered a common procedure for assisted reproduction. As the largest cells in the human body, oocytes have a large volume-to-surface ratio and a relatively high water content [8,9]. Accordingly, oocyte cryopreservation requires a very complex dehydration procedure, compared to other cells [10]. The risk of damage is relatively high during freezing and thawing as the chromosomes and meiotic spindles are exposed in the cytoplasm [11,12]. These physical and morphological characteristics make oocytes extremely sensitive to various insults during freezing and thawing [13]. To effectively cryopreserve the oocytes, a method that minimizes the formation of ice crystals is necessary [14,15]. Vitrification is a rapid cooling method that allows solidification to a glass-like state. The use of a high concentration of cryoprotectant during vitrification allows the dehydration of cells and prevents the formation of ice crystals [16,17]. Thus, vitrification is currently the recommended technique to cryopreserve embryos [18] and oocytes [19]. 

Vitrification can be performed using an open carrier system (OC) or a closed carrier system (CC) [20]. A notable advantage of OC is its excellent vitrification efficiency owing to a rapid cooling rate upon direct contact with liquid nitrogen (LN_2_) [14]. Nonetheless, the direct contact with LN_2_ may increase the risk of cross-contamination between materials from different patients, such as microbes and viruses [21], especially during the current COVID-19 pandemic. The CC has a relatively lower cooling rate and prevents cross-contamination, as the oocytes do not come into contact with LN_2_ [14,20]. Although a few reports suggest that the risk of cross-contamination is negligible [14,22], vitrification using OC remains forbidden in a few countries (e.g., France, Belgium, Ireland, and Czech Republic) [16]. The European Society of Human Reproduction and Embryology (ESHRE) and the American Society of Reproductive Medicine (ASRM) highlighted that using a high-security closed device is the safest fertility preservation procedure [23,24]. However, the advantages of both carrier systems are still debatable. Typically, OC is known to be more efficient for oocyte vitrification [25]. Thus, most laboratories prefer using open devices for both cooling and storage in LN_2_ [14,25]. Live birth rates tend to be lower when vitrification of human blastocysts and oocytes is performed using CC compared to OC [20,26]. The survival rates of human mature oocytes after vitrification using CC were relatively lower than those of OC [27]. Therefore, further studies remain necessary to balance the efficiency and safety of oocyte vitrification using CC. The purpose of this study was to improve the efficiency of conventional CC for oocyte vitrification and optimize a protocol using CC by modifying key parameters of vitrification.

## 2. Materials and Methods

### 2.1. Mouse Oocyte Collection

All experiments performed in this study were approved by the animal care and use committee of CHA University (approval no. IACUC190002). All mice were housed under temperature- and light-controlled conditions (12 h light/dark cycle) and fed *ad libitum*. Female mice (7 weeks old, B6D2F1 strain, Orient Bio, Gyeonggi, Korea) were super-ovulated by intraperitoneal injection of 5 IU pregnant mare serum gonadotropin (PMSG, Sigma-Aldrich, MO, USA) followed by an injection of 5 IU human chorionic gonadotropin (hCG) (Sigma-Aldrich) 48 h later. The mice were then sacrificed 14 h later for oocyte collection. Mature MII oocytes were obtained from the oviduct ampulla, followed by a short exposure to 0.1% hyaluronidase. Among the collected oocytes, only those with a single polar body and without cytoplasmic inclusions were used for the experiments.

### 2.2. Vitrification of Oocytes Using Different Carrier Systems

In the OC, oocytes were picked up with a fine micropipette and quickly deposited on the EM grid, which was immediately immersed in a cryovial prefilled with LN_2_. In the CC, oocytes were loaded onto the cutoff end of the microcapillary tube, and then placed in the gutter at the open end of the tube in a mini drop amount of fluid (<0.3 µL). The capillary tube was then inserted into a High-Security Vitrification (HSV) straw. The open end of the straw was closed using a heat sealer (Cryo Bio System, Paris, France). The straw was then plunged into the LN_2_.

### 2.3. Vitrification and Warming of Mouse Oocytes

Our vitrification procedure followed the methods described in a previous report (Figure 1A) [28]. Briefly, the basic solution was a mixture of HEPES buffer (SAGE Media, Trumbull, CT, USA) and 20% human serum albumin (HSA). The equilibration solution (V1) was prepared by mixing 7.5% ethylene glycol (EG) and 7.5% dimethyl sulfoxide (DMSO). The vitrification solution (V2) was composed of 15% EG, 15% DMSO, and 0.5 M sucrose. The following process was performed on a warm plate at 37 °C. Oocytes were incubated in the V1 solution for 2.5 min, and then transferred to the V2 solution. After 20 s of exposure to V2, oocytes were loaded onto EM grids (IGG 400, Pelco International, Westchester, PA, USA) or HSV (Cryo Bio System) and directly submerged in LN_2_. Vitrified oocytes were kept in the LN_2_ tank for 2–8 weeks. The cooling rate was increased by slushing LN_2_ using a vit-Master (IMT, Ness ZIona, Israel). Slush LN_2_ was applied only to CC. The experimental design for modifications during the procedure was summarized (Figure 1B).

A four-step warming process was performed. Oocyte-loaded devices were immediately plunged into 0.5 M sucrose solution for 2.5 min. Warmed oocytes were sequentially incubated in descending concentrations of sucrose (0.25, 0.125 and 0 M) for 2.5 min each. After warming, the oocytes were washed in human tubal fluid (HTF) culture medium three times and cultured for 2 h in a 37 °C incubator under at 6% CO_2_.

### 2.4. Evaluation of Oocyte Viability Following Vitrification and Warming

The morphological evaluation of vitrified/warmed (VW) oocytes was performed 2 h after the warming process. Oocytes were defined as fully surviving if intact cells were observed under a microscope. Oocytes with a spherical and symmetrical shape, homogeneous cytoplasm, intact zona pellucida and cell membrane, no sign of lysis, no membrane damage, no swelling, and no degeneration or leakage of the cellular content were considered viable.

### 2.5. In Vitro Fertilization and Embryo Culture

Sperm was collected from the cauda epididymis of 12-week-old male mice. Mature MII oocytes in HTF containing 10% KnockOut^TM^ serum replacement (Thermo Fisher Scientific, Fair Lawn, USA) were inseminated with capacitated spermatozoa (1–2 × 10^6^/mL) and incubated for 6 h. Fertilized oocytes were transferred to COOK medium (COOK, Queensland, Australia), covered with mineral oil, and incubated at 37 °C in a humidified 6% CO_2_ atmosphere. The cleavage rate was recorded on day 2, and the number of embryos that reached the blastocyst stage (hatching and/or hatched blastocyst) was assessed on day 5.

### 2.6. Immunofluorescence Staining of Oocytes and Blastocysts

For oocyte immunofluorescence staining, the zona pellucida (ZP) of the oocytes was removed using an acid Tyrode solution (pH 2.5; Sigma-Aldrich). Oocytes were then transferred to 5% normal goat serum for 2–4 h at 4 °C, and then incubated overnight at 4 °C with primary antibodies, anti-α-tubulin (1:100, ab15246, Abcam, Cambridge, UK) and anti-pericentrin (1:100, 611203, BD Biosciences, San Jose, CA, USA). After three washes in phosphate-buffered saline (PBS), the oocytes were incubated with Alexa Fluor^®^ 488 and Alexa Fluor^®^ 594 conjugated secondary antibodies (1:1000, Invitrogen, CA, USA) for 1 h at room temperature. Nuclei were stained with 4′,6-diamidino-2-phenylindole (DAPI, 1:1000, 62248, Thermo Fisher Scientific). Oocytes were mounted on a glass-bottomed dish in PBS and imaged using a Zeiss Axiovert 200M fluorescence microscope with Apotome (Carl Zeiss, Oberkochen, Germany).

For blastocyst staining, 5 days after IVF, ZP-free blastocysts were fixed in 3.7% paraformaldehyde for 30 min at 4 °C and permeabilized with 0.1% Triton X-100. For cell counting cell, the blastocysts of all groups were first stained with CDX2 (Trophectoderm (TE) cell marker, Santa Cruz, TX, USA) and E-cadherin antibodies (3195S, Cell Signaling Technology, MA, USA), and were then counterstained with DAPI (Dako, Carpinteria, CA, USA) to label the nuclei. The number of inner cell mass (ICM) cells was calculated as the total cell number minus the TE cell number.

### 2.7. Data Analysis

Each experiment was repeated at least three times. Statistical analyses were performed using the Statistical Package for the Social Sciences (SPSS 22.0; SPSS Inc., Chicago, IL, USA). Data are shown as the mean ± SEM. Data were analyzed using one-way ANOVA, followed by Duncan’s multiple range test for statistical evaluation. Results were considered to be statistically significant at *p* < 0.05.

## 3. Results

### 3.1. Comparison of the Survival and Developmental Competence of VW Mouse Oocytes in CC and OC

There was a significant difference between OC and CC in all outcomes of vitrification of mouse oocytes (Figure 2A). The survival rate of VW mouse oocytes using the CC was significantly lower than that using the OC (96.8% vs. 76.1%, *p* < 0.05) (Figure 2B). Whereas the fertilization rates of VW oocytes using the OC (91.7%) were similar to those of fresh oocytes (89.5%), they were significantly reduced in the CC (76.3%, *p* < 0.05) (Figure 2C). Furthermore, the blastocyst formation rates by the number of fertilized oocytes (81.8% vs. 72.2%, *p* < 0.05) and survived oocytes (73.2% vs. 55.1%, *p* < 0.05) were significantly reduced in VW oocytes using the CC compared to those of OC (Figure 2D,E). These results clearly demonstrated that CC lowers the survival and developmental rates of VW oocytes, suggesting that CC procedures need to be modified to be more efficient. 

### 3.2. Increased Cooling Rate Significantly Improves the Survival and Developmental Competence of VW Oocytes Using the CC

To examine the effect of increasing the cooling rate, we used slush LN_2_ (CC-sLN_2_) rather than LN_2_. The survival rate of CC-sLN_2_ was significantly higher than that of CC (85.6% vs. 73.2%, *p* < 0.05) (Figure 3A). The fertilization rate of CC-sLN_2_ was also significantly higher than that of CC (83.1% vs. 73.1%, *p* < 0.05) (Figure 3B). CC-sLN_2_ and CC showed similar blastocyst formation rates (75.0% vs. 65.8%) according to the number of fertilized oocytes (Figure 3C). However, when assessed by the number of survived oocytes, CC-sLN_2_ resulted in significantly better outcomes (62.3% vs. 48.1%, *p* < 0.05) (Figure 3D). These results collectively indicated that increasing the cooling rate by applying slush LN_2_ to the CC leads to improved survival and blastocyst formation rates of VW oocytes.

### 3.3. Increase in the Volume of First Warming Solution Improved the Survival and Developmental Competence of VW Oocytes

The volume of the first warming solution was expanded from 1 mL (CC) to 3.5 mL (CC-IV) to increase the warming rate. The survival rate of the CC-IV was significantly higher than that of CC (86.0% vs. 76.6%, *p* < 0.05) (Figure 4A). The fertilization rate of CC-IV was also significantly improved (81.5% vs. 72.9%, *p* < 0.05) (Figure 4B). Whereas the survival and fertilization rates of CC-IV were significantly higher than those of the CC, CC-IV did not improve the blastocyst formation rates (Figure 4C,D).

### 3.4. Increased Sucrose Concentration and l-Proline Supplementation Did Not Provide any Positive Effects on the Reproductive Outcomes of VW Oocytes

When the concentration of sucrose as the non-permeable protective agent in the CC (0.5 M) was increased to 1.0 M (CC-IS), the rates of survival, fertilization, and blastocyst formation of VW oocytes were comparable between the CC and CC-IS (Appendix A). When l-proline was added as a supplement (CC-LP) to the cryoprotective agent (CPA), there was no significant difference between the CC and CC-LP with respect to the rates of survival, fertilization, and blastocyst formation of VW oocytes (Appendix A).

### 3.5. Optimization of CPA Exposure Time Improved the Reproductive Outcomes of VW Oocytes

When the CPA exposure time was changed from 2.5 min (CC) to 5.0, 7.5 and 10.0 min (CC-T5.0, CC-T7.5, and CC-T10.0, respectively) in the equilibrium solution, the outcomes of VW oocytes were evaluated. The survival rates of all CC-Ts were significantly higher than that of CC (75.0%) (*p* < 0.05), with the highest rate in the CC-T10.0 (91.3%) (Figure 5A). Whereas the fertilization rate of the CC-T5.0 (80.9%) was similar to that of the CC (75.0%), those of CC-T7.5 (87.0%) and CC-T10.0 (61.6%) were significantly higher and lower than that of CC, respectively (Figure 5B). Although CC-T10.0 had the highest survival rate among all groups, it resulted in significantly low fertilization and blastocyst formation rates (Figure 5C,D). When the blastocyst formation rates were compared across all groups, CC-T10.0 was significantly detrimental to blastocyst development (26.7%) (Figure 5C). When evaluated by the number of survived oocytes, CC-T7.5 (66.7%) and CC-T10.0 (16.4%) showed significantly higher and lower blastocyst formation rates than that of CC (53.3%), respectively (Figure 5D).

### 3.6. Modified Vitrification/Warming Using CC Significantly Improved the Reproductive Outcomes of the VW Oocytes

To improve the efficiency of the conventional CC to the levels of the OC, the parameters with improved outcomes were applied in the modified vitrification/warming using CC (mVW-CC) (Figure 6A). In survival rates, the mVW-CC (96.3%) reached a comparable level to the OC (98.3%) (Figure 6B). In fertilization rates, mVW-CC (88.3%) also provided similar efficiency to the OC (91.5%) (Figure 6C). Regarding the blastocyst formation rates (Figure 6D,E), mVW-CC was also comparable to the OC. The rates of survival, fertilization, and blastocyst formation of VW oocytes using the mVW-CC were significantly increased compared to those using CC (*p* < 0.05). These results indicate that modifying the conventional CC could improve oocyte survival and developmental competence after vitrification to levels comparable to those of the OC.

### 3.7. The mVW-CC Is as Efficient as the OC in Maintaining the Structural Integrity and Developmental Competence of Oocytes after Vitrification

To examine whether the mVW-CC was also effective at the cellular levels, we analyzed the spindle morphology of VW oocytes and the number of total cells in the blastocysts developed from the VW oocytes. The percentage of the oocytes with normal meiotic spindle assembly in the mVW-CC (15/20, 75.0%) was significantly improved compared to that of the CC (23/51, 45.1%) and comparable to that of the OC (35/47, 74.5%) (Figure 7A,B). In the total cell number of the blastocyst developed from VW oocytes, the OC, CC, and mVW-CC showed a mean of 74 ± 7.9, 57 ± 17.7, and 77 ± 15.0, respectively (Figure 7D). In the number of TE, the OC, CC, and mVW-CC showed a mean of 60 ± 7.3, 43 ± 16.9, and 63 ± 15.0, respectively (Figure 7E). In the number of ICM, the OC, CC, and mVW-CC showed a mean of 14 ± 1.1, 14 ± 1.6, and 14 ± 2.0, respectively (Figure 7F). Moreover, the OC, CC, and mVW-CC showed mean ICM/TE ratios of 0.23 ± 0.5, 0.38 ± 0.13, and 0.24 ± 0.7, respectively (Figure 7G). In the blastocysts developed from VW oocytes, the total cell numbers, TE cell numbers, and ICM/TE ratios in the mVW-CC were significantly higher than those in the CC (*p* < 0.05), reaching the levels of the OC (Figure 7C).

## 4. Discussion

Efficient vitrification of mammalian oocytes is of great importance in assisted reproductive technology (ART). However, the vitrification of mammalian oocytes using CC has a low success rate. Thus, we modified four key parameters, namely, thermodynamic characteristics (cooling and warming rates), CPA exposure time, and CPA supplementation, to optimize the oocyte vitrification protocol using the CC. Most laboratories use OCs for both cooling and storage in LN_2_ during vitrification. There is a reluctance to use the CCs because of the hypothetical reduction in the cooling rate [25], although CCs have the advantage of not being physically exposed to LN_2._ When plunged in LN_2_, more boiling occurs as a result of heat exchange due to the air between the outer cap sleeve and inner straw in the CCs [29]. This can cause mitochondrial degeneration (membrane disintegration, matrix fragmentation, cristae disappearance) and microvilli deformation, thereby damaging the oocyte [30]. Increasing the cooling rate by using slush LN_2_ in the CC overcame these drawbacks and improved the survival rate and developmental competence of VW oocytes (Figure 3). Slush LN_2_ improves the efficiency of OCs in the vitrification of human oocytes [31,32], human ovarian tissue [33,34], and mouse oocytes [35]. It reduces chilling injury and re-crystallization [36], and is effective in maintaining meiotic spindles in VW human oocytes using OCs [32].

Recently, there has been a change in the paradigm of vitrification. Several studies have reported that an increase in the warming rate is more critical to achieve a higher survival rate than increasing the cooling rate [37,38,39]. Various methods, such as laser pulses [37], magnetic induction heating nanoparticles [38], and ultrarapid warming [39] have been used to increase the warming rate in the vitrification of mouse oocytes, embryos, and/or mesenchymal stem cells. In this study, we increased the volume of the first warming solution to accelerate the warming rate, which significantly improved the survival and fertilization rates of VW oocytes using CC (Figure 4). These data are supported by a previous report that showed an increase in the volume to 3.5 mL for equine germinal vesicle oocytes using the OC, resulting in improved survival and developmental rates [40]. However, increasing the volume of the first warming solution did not increase the blastocyst formation rates of the VW oocytes after fertilization, suggesting that further optimization is required. Thus, increasing both the cooling and the warming rates seems to be effective. Our mVW-CC with the increase in both cooling and warming rates improved the lower efficiency of the CC without risks of potential contamination to the VW oocytes (Figure 6). Additional studies are required to identify optimized conditions with good efficacy and safety for oocyte vitrification [14,25].

It is known that increasing viscosity can reduce the formation of ice crystals and protect cells from strong osmotic shock during warming. Sucrose is the most commonly used nonpermeable CPA in vitrification/warming solutions [41]. The use of 0.5 M nonpermeable CPA during warming is considered sufficient. To improve the survival rate of VW oocytes, many studies have attempted to stabilize the rehydration process during warming by increasing sucrose concentrations [27,42]. However, increasing the viscosity of the warming solution did not show any positive effects on the survival and embryo development of VW oocytes using the CC in this study (Appendix A). Previous studies have also shown that the survival and cleavage rates are not significantly different between 1.0 M and 0.5 M sucrose solutions in bovine oocytes [43,44]. In contrast, sucrose concentrations higher than 1 M may have detrimental effects on VW oocytes, as observed in mouse ovarian tissues [45]. A recent review on nonpermeable CPAs suggested that 0.5–1.0 M sucrose is applicable [41]. l-Proline can act as a natural osmo-protectant without increasing the CPA concentration. Although l-proline supplementation did not provide any positive outcomes to VW oocytes using the CC (Appendix A), it was reported to improve the survival rate and mitochondrial activity of VW mouse oocytes using the OC [46]. Further studies are warranted to determine whether higher viscosity with l-proline may stabilize spindle assembly and mitochondrial activity in VW oocytes using mVW-CC.

Since CPA toxicity is time- and concentration-dependent, the composition of CPA solutions and the duration of the exposure time are critical [47]. In blastocyst vitrification using the OC, changing the CPA exposure time did not affect the clinical and neonatal outcomes of human blastocysts [48], while increasing the CPA exposure time improved DNA integrity in mouse blastocysts [49]. Due to the reduced cytotoxicity and osmotic stress, short-term equilibration time improved the re-expansion and hatching rates [50] and increased the total cell number with a reduced mRNA expression of apoptosis-related genes, such as *Bax* and *Bcl2l1*, in bovine blastocysts [51]. Although a shorter CPA exposure has been proposed to reduce CPA cytotoxicity [31,32,52,53], other studies have recommended a longer exposure time for human oocytes using OCs [54,55,56,57]. Whereas longer CPA exposure (>10 min) significantly increased the survival rate of VW oocytes, it severely impaired their developmental competence (Figure 5). However, some reports have also shown that an exposure time that is too short adversely affects the survival and development of VW oocytes using the OCs [58,59]. When CC was used, CPA may not be completely replaced in oocytes exposed for a short time, such as 2.5 min, resulting in damage to the organelles in the oocytes (Figure 7). The optimal exposure time to CPA in oocyte vitrification using CCs has not yet been standardized. Our results suggest that an appropriate CPA exposure time longer than 2.5 min but shorter than 10 min is recommended to minimize CPA cytotoxicity on the developmental competence of survived VW oocytes while increasing the vitrification efficiency of mouse oocytes.

Here, we demonstrate that the mVW-CC for oocyte vitrification provides better outcomes regarding the survival, fertilization, and developmental rates of the VW oocytes. In addition, the majority of VW oocytes using our mVW-CC had barrel-shaped meiotic spindles that were well aligned with the chromosomes (Figure 7A). However, this study has several limitations. While we clearly showed that our mVW-CC improved the reproductive outcomes of the VW oocytes in vitro, the neonatal outcomes of the embryos developed from the VW oocytes are lacking. Our results cannot be applied to other mammals because oocytes of other species may have different biochemical and biophysical properties that require different conditions. Further studies are strongly warranted to apply it to human oocytes.

The mVW-CC method may help reduce the cytotoxicity and osmotic pressure during the vitrification/warming procedure. Further studies are strongly warranted to decipher the molecular mechanisms underlying the stabilization of oocyte integrity after and/or during the vitrification/warming processes. In summary, this study shows that the efficiency of oocyte vitrification using CC can be improved by increasing the cooling/warming rate and CPA exposure time (Figure 8). This study may contribute to the improvement of the efficiency and safety of vitrification using CC in human oocytes.

## Figures and Tables

**Figure 1 cells-10-01670-f001:**
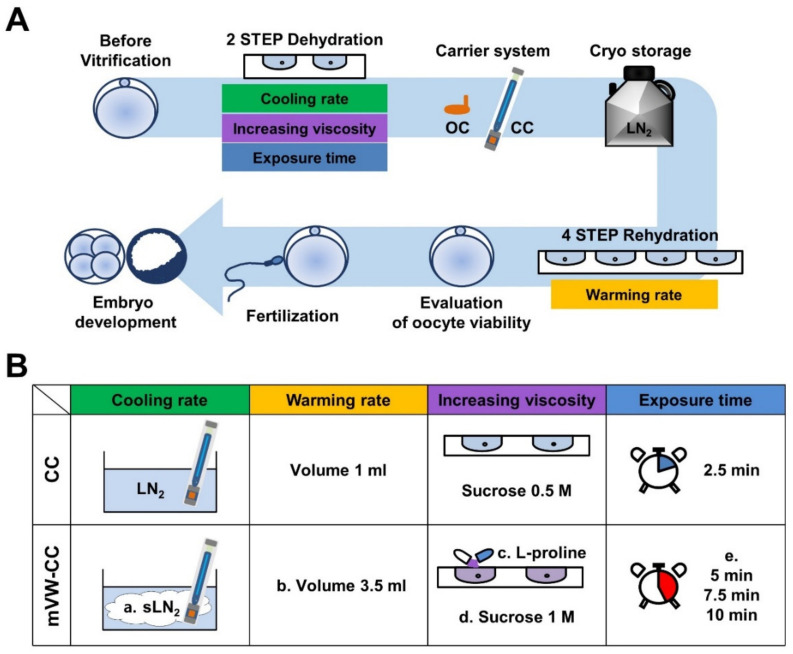
Schematic representation of the vitrification and warming procedure using mouse oocytes. (**A**) During vitrification, mature oocytes underwent a two-step dehydration cooling process, and then the oocytes were loaded on two different carrier devices, an open carrier system and a closed carrier system, and immersed in liquid nitrogen. Oocytes were warmed via a stepwise transfer into four sucrose concentration solutions, 0.5, 0.25, 0.125 and 0 M for 2.5 min each. After 2 h of incubation in fertilization medium, oocytes were subjected to insemination. (**B**) The experimental design for four different modifications during the procedure: (a) increased cooling rate by slush liquid nitrogen; (b) increased warming rate by expanding the volume of first warming solution; (c,d) Increased viscosity by supplementation of l-proline and increased sucrose concentration; (e) increase in exposure time to cryoprotective agent.

**Figure 2 cells-10-01670-f002:**
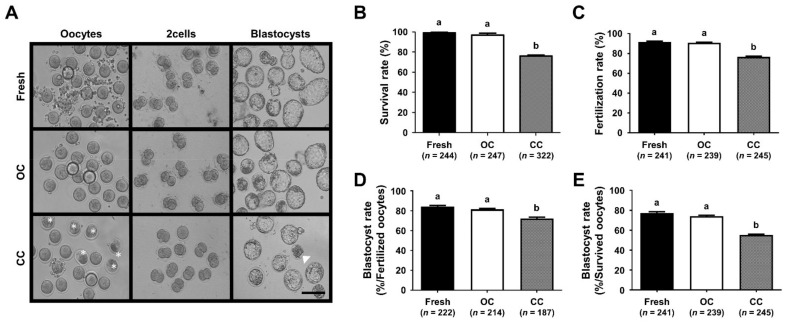
Comparative analyses of reproductive outcomes of vitrified/warmed (VW) oocytes using two different carrier systems. (**A**) Representative microscopic images of pre-implantation embryos in vitro fertilized and developed from VW oocytes using different carrier systems. White asterisks and an arrowhead indicate lysed oocytes and arrested embryo, respectively. Scale bar = 140 µm. (**B**–**E**) Comparison of the survival (**B**), fertilization (**C**), and blastocyst formation rates (**D**,**E**) of VW oocytes with different carrier systems. Blastocyst formation rates (the number of blastocysts/the number of fertilized oocytes (**D**) and/the number of survived oocytes (**E**)). Different superscript letters (a, b) indicate *p* < 0.05. The number of experiments = 10 (24–33 oocytes per each group in each experiment). Fresh, fresh control; OC, open carrier system; CC, closed carrier system.

**Figure 3 cells-10-01670-f003:**
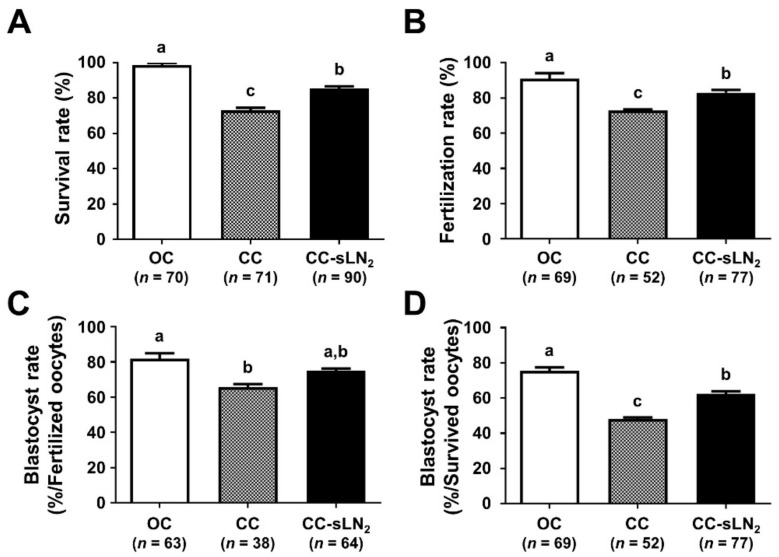
The effects of slush liquid nitrogen (LN_2_) used to increase the cooling rate on the survival and development rates of vitrified/warmed (VW) oocytes. Comparison of the survival (**A**), fertilization (**B**), and blastocyst formation rates (**C**,**D**) of VW oocytes exposed to slush LN_2_. Blastocyst formation rates (the number of blastocysts/the number of fertilized oocytes (**C**) and/the number of survived oocytes (**D**)). Different superscript letters (a–c) indicate *p* < 0.05. The number of experiments = 5 (13–19 oocytes per group in each experiment). OC, open carrier system; CC, closed carrier system with LN_2_; CC-sLN_2_, closed carrier system with slush LN_2_.

**Figure 4 cells-10-01670-f004:**
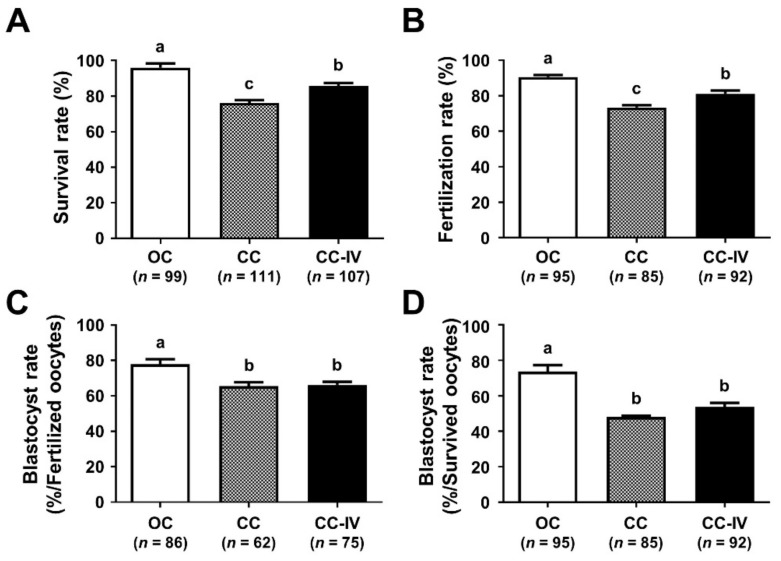
The effects of expanding the volume of the first warming solution to increase the warming rate on the survival and development rates of vitrified/warmed (VW) oocytes. Comparison of the survival (**A**), fertilization (**B**), and blastocyst formation rates (**C**,**D**) of VW oocytes exposed to enlarged volume of first warming solution. Blastocyst formation rates (the number of blastocysts/the number of fertilized embryos (**C**) and/the number of survived oocytes (**D**)). Different superscript letters (a–c) indicate *p* < 0.05. The number of experiments = 4 (20–25 oocytes per group in each experiment). OC, open carrier system; CC, closed carrier system with standard volume (1 mL); CC-IV, closed carrier system with an increase in the volume of first warming solution (3.5 mL).

**Figure 5 cells-10-01670-f005:**
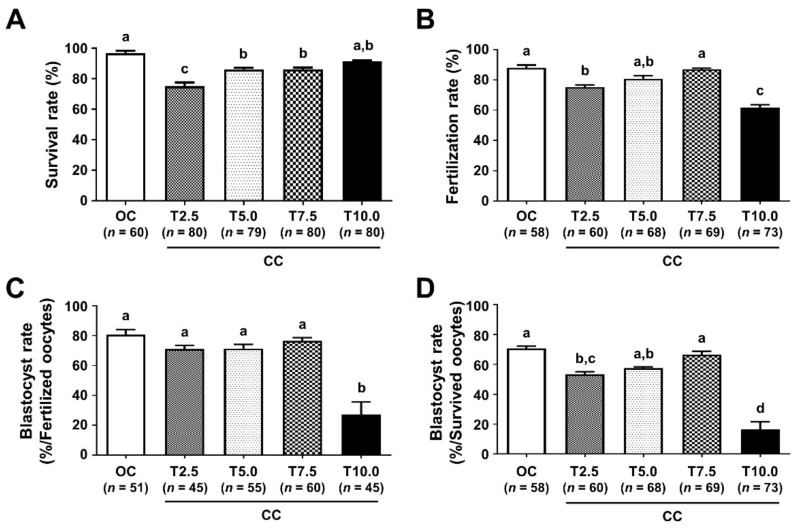
Comparison of the survival and development rates of vitrified/warmed (VW) oocytes with different CPA exposure times. Comparison of the survival (**A**), fertilization (**B**), and blastocyst formation rates (**C**,**D**) of VW oocytes with different carrier systems. Blastocyst formation rates (the number of blastocysts/the number of fertilized oocytes (**C**) and/the number of survived oocytes (**D**)). Different superscript letters (a–d) indicate *p* < 0.05. The number of experiments = 3 (23–28 oocytes per group in each experiment). OC, open carrier system; CC, standard exposure time (2.5 min, CC-T2.5) to equilibrium solution in closed carrier system; CC-T5.0, CC-T7.5, and CC-T10.0, exposure time increased to 5.0, 7.5 and 10.0 min in the equilibrium solution, respectively.

**Figure 6 cells-10-01670-f006:**
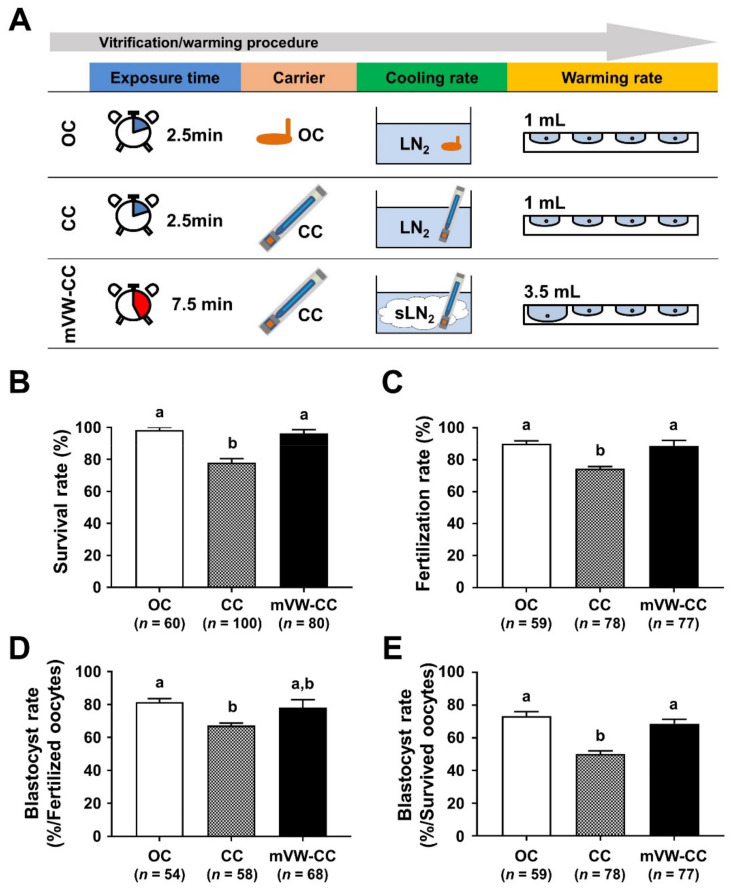
Comparative analyses of the survival and development rates of oocytes vitrified/warmed (VW) using open carrier system (OC), closed carrier system (CC), and modified vitrification/warming CC system (mVW-CC). Comparison of the procedure (**A**), survival (**B**), fertilization (**C**), and blastocyst formation rates (**D**,**E**) of VW oocytes using different carrier systems. Blastocyst formation rates (the number of blastocysts/the number of fertilized oocytes (**D**) and/the number of survived oocytes (**E**)). Different superscript letters (a,b) indicate *p* < 0.05. The number of experiments = 3 (18–35 oocytes per group in each experiment).

**Figure 7 cells-10-01670-f007:**
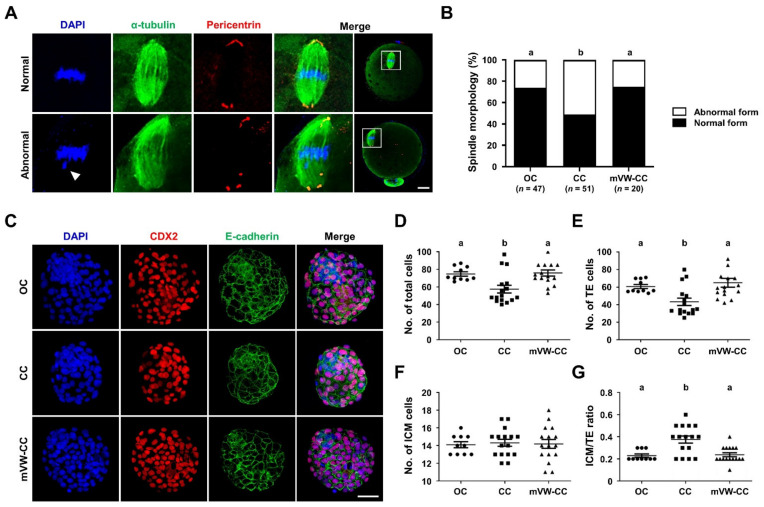
Morphological and functional assays show that the efficacy of vitrified/warned (VW) oocytes in the modified vitrification/warming closed carrier system (mVW-CC) is identical to that in the open carrier system (OC). (**A**) Representative images of immunofluorescence staining for α-tubulin (spindle marker), pericentrin (spindle pole marker), and DAPI (nuclei) in VW oocytes with either normal (upper panels) or abnormal (down panels) meiotic spindle organization and chromosomal alignment. The arrowhead indicates an abnormally aligned chromosome. Scale bar = 20 µm. (**B**) Quantitative analyses of VW oocytes with normal spindle and chromosome morphologies. White and black bars represent the percentages of oocytes with abnormal and normal spindles, respectively. (**C**–**G**) Analyses of the blastocyst developed from VW oocytes using mVW-CC. (**C**) Representative images of immunofluorescence staining for CDX2 (trophectoderm (TE) marker) with DAPI in the blastocysts developed from VW oocytes using different vitrification methods. (**D**–**G**) Quantitative analyses of the total cell number, TE cells, inner cell mass (ICM), and ICM/TE ratio in the blastocysts were done on the basis of the number of DAPI-positive, CDX2-positive, and DAPI-positive and CDX2-negative cells, respectively. The horizontal black lines represent the median values of the total, TE, and ICM cell numbers and ICM/TE ratios. Scale bar = 50 µm. Different superscript letters (a,b) indicate a significant difference at *p* < 0.05. The number of oocytes and blastocysts examined = 20–51 (**A**,**B**) and 10–16 (**C**–**G**), respectively, in each group. CC, closed carrier system.

**Figure 8 cells-10-01670-f008:**
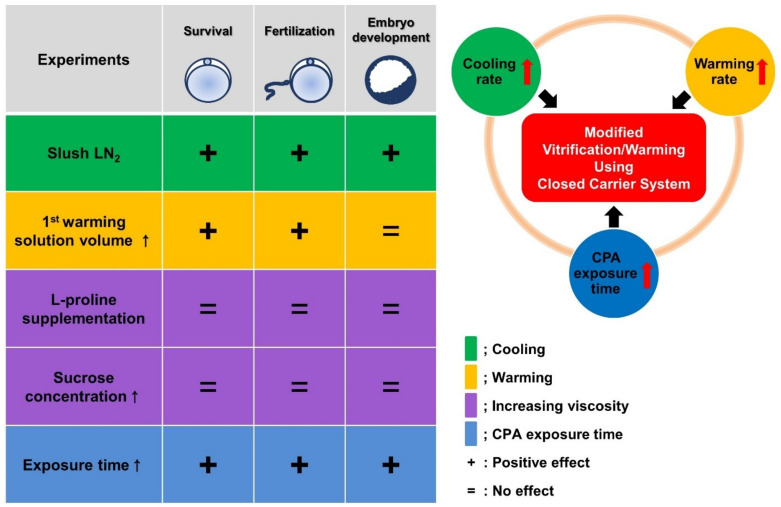
Schematic illustrations that summarize the outcome of the modified key parameters in the conventional closed carrier system (CC). Each color indicates a parameter that was modified, and ↑, +, and = represent an increase, a positive effect, and no effect, respectively.

## Data Availability

Not applicable.

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
