# Peer review of "Development of Optimized Vitrification Procedures Using Closed Carrier System to Improve the Survival and Developmental Competence of Vitrified Mouse Oocytes"

_cells, 2021, doi:10.3390/cells10071670_

Round 1
Reviewer 1 Report
I accept all changes made to the manuscript. I have no more comments.
Author Response
Reviewer 1's report: I accept all changes made to the manuscript. I have no more comments.
We really appreciate the reviewer's acceptance of our manuscript.
Reviewer 2 Report
Please see the attached file.

Round 2
Reviewer 2 Report
The authors modified the article according with reviewer's suggestions
This manuscript is a resubmission of an earlier submission. The following is a list of the peer review reports and author responses from that submission.
Round 1
Reviewer 1 Report
The paper is very interesting. So far vitrification of oocytes in the CC system was significantly lower compared to OC system. Optimization of the vitrification process by the mCC system is safer for oocytes or embryos by decreasing the risk of cell contamination by bacteria or viruses. The modification in cooling, warming, CPA exposure time, viscosity enlarge percentages of survival oocyte and blastocyst rate. The authors also described the limitations of this research e.g. lack of birth rate or analysis only on mice cells. The results in mice are very promising and required further investigations in women or females of breeding animals. The concept, methodology and research results are adequately presented. The manuscript could be recommended for publication, but needed some modifications.
Minor comments:
Abstract : CPA- explain abbreviation
Line 116-120: Figure 1b- the description is about four studies group (cooling, warming, viscosity, exposure time), but the Authors wrote about the “procedure of present six groups”. This is confusing.
Figure S1, S2- lack of OC position (open carrier system) on graphs, although is in description of figure.
Line 136-157: Any control staining was performed for oocytes?
Figure 6. Please name on graph “standard exposure time”, e.g. T2.5 or ST. The description of this figure is not clear.
Figure 4-6. Legend: The result from OC is missing. All figures 4-6 (A-D) show only results in the CC system.
During the IVM, embryo-transfer procedure the quality of oocytes and then blastocyst are very important. Maybe the Authors may estimate, how a percentage of good quality blastocysts was achieved in OC vs CC or mCC system. Are Figures 2-4 show only good quality blastocyst rate (or also with some degenerative changes)?
Reviewer 2 Report
The purpose of this study was to assess various solutions to increase survival and clinical performance when using Closed Carriers (HSV by CBS) for mouse oocyte vitrification.
The study was performed on mouse oocytes.
Major comments:
Despite the effort of the authors this article is very confused and hard to be read. The authors are proposing various solutions to increase survival rate when using the Closed System. They summarized a combination of solution in the vague acronym of modified closed carrier mCC.
The so-called mCC is a mixture of modification in the way to use the carrier (pre-warming of external straw), in vitrification procedure (changes in equilibration), in viscosity of vitrification solution and in the volume of the warming solution, and last but not least the use of slush nitrogen.
Based on authors’ conclusions, these adjustments seem to improve the performance of HSV. Any adjustment was compared with the CC, it is not clear if all the adjustments together were tested in the same moment.
The frequencies do not seem available in the text. As far as I can see, I can read only percentages. This article cannot be published without these numbers. If frequencies are present in the article, probably they are not easily readable, and the authors must highlight these with tables.
Page 5 line 183: the pre-cooling of the external straw transforms the closed carrier in a semi-closed carrier (or supercooled air/vapor phase vitrification). With this method the contamination during the vitrification procedure is virtually the same that with open carrier. This matter has been extensively discussed by various authors (Larman and Gardner, Parmegiani, Vajta, ect..). Regarding this aspect… when using pre-cooled air, the HSV cannot be more considered as a Closed Carrier. Just this invalidates the study. This modify makes HVS an semi-closed or open carrier (it is just a matter of nomenclature) , because vitrification occurs due to the contact with Nitrogen vapour, which can be contaminated (see the literature on supercooled air vitrification and carrier nomenclature) .
Page 6 line 200. Slush nitrogen is a modify for the procedure not for the carrier.
Page 7 line 220. The increase of warming volume is a modify for the procedure, not for the Carrier
Minor comments:
Page 2 lines 60-62 the cryostorage is not directly related with the vitrification/warming procedure. A carrier can be open for vitrification and closed for cryostorage. So, the guidelines cited are not referring the vitrification procedure but the cryostorage.
Page 3 line 98: In humans, oocytes equilibration step is 15 minutes. If this article would like to suggest anything for humans probably the equilibration duration proposed in this study (2,5 minutes) should be revised or discussed.